# Initial Effect of Recombinant Human Growth Hormone Treatment in a Patient with Löwe Syndrome

**DOI:** 10.3390/children10071166

**Published:** 2023-07-05

**Authors:** Violeta Iotova, Teodora Karamfilova, Mariya Levkova, Mariya Gaydarova, Sonya Galcheva, Dimitrichka Bliznakova

**Affiliations:** 1Department of Pediatrics, Medical University Varna, 9000 Varna, Bulgaria; 2Department of Medical Genetics, Medical University Varna, 9000 Varna, Bulgaria; 3Department of Pediatrics Nephrology and Dialysis, Medical University Sofia, 1000 Sofia, Bulgaria

**Keywords:** Löwe syndrome, impaired growth, growth hormone treatment

## Abstract

Objectives: Löwe syndrome (the oculocerebrorenal syndrome of Löwe, OCRL, OMIM #309000, ORPHA: 534) is a very rare multisystem X-linked disorder characterized by ocular, kidney and nervous system anomalies. Case presentation: We present the first Bulgarian genetically confirmed patient with OCRL. The patient had facial dysmorphism, cryptorchidism, congenital cataracts, nystagmus, delayed physical and mental development, and poor nutritional status. He had severe rickets, metabolic acidosis, hypokalaemia, hypophosphataemia, and low IGF-1 levels at the age of three, in addition to his developmental delay. The molecular-genetic analysis reported a pathogenic variant c.1124A>G, p.H375R in the *OCRL* gene. This variant was inherited from the mother, who was a carrier. Following the diagnosis of OCRL, treatment with potassium citrate, phosphate, and calcitriol was initiated, along with an increase in caloric intake. Following general physical and biochemical improvement, therapy with rhGH started 4 years ago, and current results are presented. Conclusions: The patient with Löwe syndrome who was presented with a 6-year follow-up demonstrates the complexity of rare disease cases and the value of multidisciplinary care together with growth hormone treatment for better results in these patients.

## 1. Introduction

Löwe syndrome (oculocerebrorenal syndrome, ORPHA:534, OMIM#309000) is a very rare hereditary disorder characterized by X-linked inheritance [1]. Congenital cataracts, mild to severe intellectual disability, and renal tubular malfunction that results in slowly progressing renal failure are its three distinctive features [2]. Renal Fanconi syndrome may be present in the first postnatal months and differ in severity between individuals. It may sometimes be asymptomatic or have an unusual clinical presentation [3].

This disorder occurs almost exclusively in males, while females are usually non-symptomatic carriers [2]. Pale complexion, growth retardation, intellectual disability, areflexia, hypotonia, convulsions, and non-inflammatory joint swelling are common symptoms. Most of these symptoms are due to the generalized dysfunction of the proximal tubule of the kidney, which results in excessive excretion of water, bicarbonates, glucose, and amino acids [1]. Pathogenic variants in the *OCRL* gene on chromosome Xq25-26 cause this disorder. The *OCRL* gene codes for the protein OCRL-1, which is a phosphatase enzyme [4]. This enzyme is involved in cell transport and maintaining the integrity of the cell by regulating the actin cytoskeleton [3], and its impaired function may potentially affect many different organs.

The primary differential diagnosis is Dent disease (OMIM #300009, OR-PHA:1652). Dent disease is characterized by proteinuria, hypercalciuria, nephrocalcinosis, and urolithiasis [5]. Less common features of Löwe syndrome also include rickets and mildly short stature. By early to mid-adulthood, progressive kidney issues frequently result in renal failure [3]. Variants in the *OCRL* gene were reported in some patients with Dent disease, now known as Dent disease type 2 (OMIM #300555, ORPHA:314721) [6]. The two forms of Dent disease are distinguished based on their genetic causes. Both are inherited in an X-linked recessive manner, but Dent disease type 1 is caused by a mutation in the *CLCN5* gene [5]. Some Dent-2 patients have minor extrarenal symptoms of the syndrome, indicating that this condition is a milder variant of Löwe syndrome [1].

There are hardly any published epidemiological data about the occurrence and prevalence of Löwe syndrome. The estimated prevalence is between 1 and 10 affected males per 1,000,000 inhabitants, according to the Löwe Syndrome Association (LSA) in the USA, with 190 recognized patients residing in the country in 2000 (a prevalence of 0.67 per million inhabitants) [2]. In 2005, according to the Italian Association of Löwe Syndrome, there were 34 Löwe syndrome patients in Italy (33 boys and one girl; a prevalence of 0.63 per million people) [2]. Given the low prevalence of the disorder, any new patient report is crucial in order to learn more about this rare disease.

We present the first genetically confirmed Bulgarian patient with Löwe syndrome with a 6-year follow-up. The family provided written informed consent prior to publication.

## 2. Case Description

The boy was born from a first normal pregnancy and delivery with a birth weight of 3160 g, a birth length of 51 cm, and a head circumference of 32 cm. The development within the first few postnatal weeks was normal. When the boy was 2 months old, he was diagnosed with bilateral congenital cataracts, which were operated on and replaced with lens implants. Soon after, the patient developed glaucoma. He gradually developed nystagmus and hypotonia, and his linear growth started to slow down.

The boy was admitted to a pediatric genetic department at 7 months of age. Cytogenetic analysis and metabolic screening were normal. Poor nutritional status, facial dysmorphism, cryptorchidism, and physical and mental development were all observed but not distinctively diagnosed. The patient was discharged from the hospital without a definitive diagnosis.

By the age of 3, the boy was unable to stand with and without assistance and could not walk. He had signs of spontaneous fractures. The patient did not pronounce any words. He had an average daily liquid intake of approximately 250 mL and rejected all solid food. Urine production was extremely low, and the patient had no control over urination or defecation. The boy was first seen by a pediatric endocrinologist at the age of 3 years and 7 months due to his delayed growth. He had rickets, severe muscle hypotonia, and severe intellectual delay. Cryptorchidism and poor nutritional condition were additionally noticed. Aside from the mother and maternal grandmother’s prior diagnosis of mild thalassemia, the family history was uninformative.

Based on the CDC Growth chart (www.cdc.gov/growthcharts, accessed on 31 May 2023), the patient’s weight at presentation was −7.2 SD (8.300 kg), and his height was −13.6 SD (79 cm) (Figure 1). The results of the laboratory tests revealed low insulin-like growth factor 1 (IGF-1), hypophosphatemia, hypokalemia, and metabolic acidosis (Table 1). While proteinuria, phosphaturia, and glucosuria were evident and persisted over the course of follow-up, kidney function was not compromised. 

The total 25-hydroxyvitamin D3 was low, while PTH-intact was high (Table 1). According to the Greulich and Pyle method, the bone age (BA) was severely delayed (9 months at the chronological age of 3 years, 9 months), and bone radiograms showed alterations that were consistent with severe rickets (Figure 2). There were no signs of nephrocalcinosis. Ultrasound (US) examination confirmed bilateral cryptorchidism. Pachygiria, dilated Virchow–Robin perivascular spaces, and a normal pituitary body and stalk were all seen on the magnetic resonance imaging (MRI).

The clinical diagnosis of Löwe syndrome was established based on growth retardation, intellectual disability, severe rickets, tubulopathy, hypophosphatemia, metabolic acidosis, hypokalemia, and evidence of congenital cataracts and hypotonia at birth.

Despite the fact that renal function was unaffected, chronic anemia with hemoglobin levels around 90 g/L was discovered. Hemoglobin electrophoresis was done at normal serum iron levels because the mother and maternal grandmother had mild thalassemia. It was confirmed that the patient also had mild thalassemia.

After the family gave informed consent, Löwe syndrome was genetically proven at the Mayo Clinic in the United States. Both the patient and his mother, who was a carrier of the pathogenic variant (c.1124A>G, p.H375R), had a mutation of the *OCRL* gene in a hemizygous state. Typical cortical opacities of the lens were confirmed in the mother thereafter.

Treatment with potassium citrate, oral phosphate, and calcitriol drops was initiated when the clinical diagnosis of Löwe syndrome was evident. Attempts to increase caloric and fluid intake were commenced. In roughly a year (4 years and 6 months), the patient’s physical strength and neuropsychological development improved, and his potassium, calcium, pH, bicarbonates, base excess, and wrist radiograms were all within normal ranges (Table 1). Although there was a slight catch-up in linear growth, the rate of growth remained below average, and IGF-1 levels were extremely low at 29 ng/mL (−2.47 SD, www.specialtytesting.labcorp.com/resources/tools/endocrinology-calculator, accessed on 18 July 2018).

After achieving the recommended caloric and nutritional intake, as well as increased water intake, at the age of 5 years and 6 months, height was 89 cm (−4.65 SD), weight was 13 kg (−3.78 SD), recombinant human growth hormone (rhGH) treatment was started at an initial dose of 0.023 mg/kg/d with variable growth velocity over time (Figure 1). The rhGH dosage was gradually increased and stabilized at 0.055 mg/kg/d (Table 1).

The patient is currently 9 years 6 months old and has made remarkable improvements in both his physical (he walks by himself with some hand support) and mental development (he has 10–20 words in his vocabulary). Throughout the monitoring period, kidney function remained stable (Table 1). The current BUN serum level is 6.5 mmol/L, the creatinine level is 71 mmol/L, K^+^ is 4.3 mmol/L, Ca^2+^ is 2.5 mmol/L, and pH is 7.45. Radiograms of the wrists are normal with adequately progressing BA (8 years 6 months at the chronological age of 9 years 6 months, Figure 3). Additionally, kidney US remained normal. Fluid intake increased to the normal level for body size, especially after introducing rhGH treatment.

The patient has received rhGH therapy for more than 4 years, has gained 24.9 cm (1st-year growth velocity—10 cm) and tolerates the medication extremely well. The patient weighed 21 kg (−2.2 SD) and was 113.9 cm (−3.3 SD) tall at his most recent visit. Testicles were found in the scrotum. No side effects of the rhGH were noted.

The medical history noted that the patient experienced undefined seizures in infancy that were not treated. At 8 years of age, the patient experienced a short (less than 1 min) tonic–clonic seizure. A diagnosis of symptomatic epilepsy was made following an outpatient EEG with non-specific changes. Valproic acid treatment was started without further paroxysms so far. The rhGH treatment was not interrupted, and the dose was not changed.

Accepting only mashed food of appropriate quality and quantity results in persisting nutritional issues and is one of the signs of expressed stubbornness in Löwe syndrome patients. The multidisciplinary team caring for the patient consists of a pediatric endocrinologist, nephrologist, neurologist, nutritional therapist, and ophthalmologist who follow him twice yearly. The patient only experienced a single, mild upper respiratory tract infection over the course of the whole follow-up period.

## 3. Discussion

As already discussed, patients with OCRL syndrome typically present in infancy with congenital cataracts, growth failure and intellectual disability [4]. Dense congenital bilateral cataracts are the hallmark of Löwe syndrome and are present at birth [2]. Cataracts develop early in embryogenesis [7] and have been demonstrated on prenatal ultrasound images [8]. About 50% of those with Löwe syndrome have severe glaucoma with buphthalmos that require surgical treatment, generally in the first year but occasionally as late as the second or third decade [9]. Management includes early lens extraction and the prescription of eyeglasses, while surgical lens implants are not recommended. Additionally, Loi warns against prescribing contact lenses due to the possibility of corneal keloid development [2]. Our patient had lens implants at the age of six months, mostly because the underlying disease had not been determined at that point. The ocular tone was frequently assessed, and glaucoma was then treated as appropriate. Severe newborn hypotonia, frequently without deep tendon reflexes, is notable among the early signs [2]. This symptom was also present in our patient at the age of 6 months. This calls into question whether patients with bilateral congenital cataracts should be routinely screened for additional Löwe syndrome-related symptoms.

The majority of Löwe syndrome patients have a severe intellectual disability. Up to 50% of patients experience seizures [10]. Patients with Löwe syndrome also have a distinctive pattern of behavioral anomalies. Over 80% show stubbornness, aggression, irritability, temper tantrums, and complex repetitive movements like hand flapping [1].

Based on the characteristic occulocerebrorenal symptoms, the presented patient was clinically diagnosed with Löwe syndrome. Patients with Löwe syndrome exhibit selective proximal tubule dysfunction, and the current findings are consistent with other reported cases [11]. Impaired proximal tubular reabsorption led to proteinuria, metabolic acidosis, and phosphaturia. The patient also had severe vitamin D deficiency. Usually, Löwe patients develop vitamin D-dependent rickets due to the impaired activation of 1-alpha-hydroxylase with normal 25-hydroxyvitamin D3 levels in addition to renal hypophosphatemia [3,12]. In our case, the vitamin D deficiency led to a further increase in PTH levels and severe skeletal manifestations.

Renal impairment is primarily characterized by renal Fanconi syndrome. The severity of the renal disease can vary between patients and tends to worsen with age. During the second decade of life, a significant number of patients develop chronic kidney injury, which can result in end-stage renal disease and the need for dialysis [2]. A minority of patients have been successfully treated with renal transplantation [2]. However, even in the presence of well-corrected Fanconi syndrome, some patients have repeated pathologic bone fractures with poor healing [1].

The described neuropathological findings are diverse and non-specific and may include ventriculomegaly, brain atrophy, cerebellar hypoplasia, pachygyria, polymicrogyria, abnormal neuronal migration, subependymal cysts, and cysts located in the white matter [10]. The brain MRI of our patient showed consistent changes.

The patient in our case did not exhibit nephrocalcinosis so far, despite the fact that it occurs in about 50% of patients with Löwe syndrome [1,13]. As part of the treatment, potassium citrate is helpful as it corrects both hypokalemia and metabolic acidosis, and it has been found to delay nephrocalcinosis [2]. Potassium citrate therapy was initiated for this purpose, and it has been successful up to this point. A typical finding in 33–82% of patients having Löwe syndrome is hyperchloremic metabolic acidosis [10]. Even in non-acidotic patients, the total carbon dioxide concentration in the plasma is typically detected at the lower limit of its reference range [1]. Renal tubular acidosis (RTA) appears to be less prevalent in patients with Dent-2 disease [13].

Although there is a link between renal function and growth impairment, Löwe syndrome patients exhibit severe short stature at all stages of renal impairment [14]. The current patient’s height was below the third percentile at his initial evaluation. As previously mentioned, the defining feature of Löwe syndrome is severe postnatal growth retardation, which is unrelated to the severity of renal failure or bone disease [13]. By 3 years of age, the mean height of Löwe syndrome patients falls to the third percentile, and it continues to fall with age [10].

The etiology of growth delay in children with kidney disease is generally multifactorial, including rickets, growth hormone (GH) resistance, reduced GH secretion rate or greater loss of GH, and functional IGF-1 deficiency. Both endogenous and exogenous GH lead to increased glomerular filtration rate (GFR). It is likely that the increased GFR is mediated by the increased IGF-1. There is very little research available on the use of rhGH in Löwe syndrome patients. In one paper, a prepubertal, 17-year-old Löwe syndrome patient with multiple fractures, extreme stunting, and osteopenia received a combination of intravenous Pamidronate treatment with rhGH and the standard therapy for renal Fanconi syndrome [15]. It was previously stated that the use of rhGH should be limited to patients with an evident GH deficiency [2]. In our patient, despite the better nutritional and physical status, growth velocity, bone age, and IGF-1 did not improve. That is why instead of GH testing, we decided to start a therapeutic test with rhGH in smaller doses than the recommended in chronic renal disease, and after metabolic and nutritional improvement under close surveillance of the patient, 4 years after starting rhGH, the positive aspects of treatment are evident. There is an improvement in growth, muscle tone, and general state. There is no longer a problem with cryptorchidism, and additional surgery is not anticipated. Thus, it is not surprising that in a recent series of 137 patients, Sena et al. showed that 15% of patients were favorably treated with rhGH [16].

Long-term rhGH treatment raises the risk of hyperfiltration with resultant glomerulosclerosis and an accelerated decline of renal function [17]. We cannot elaborate on the prognosis of the patient since it is very variable with age. Currently, there are no reports on the effect of rhGH treatment on a patient’s prognosis. We will follow the patient closely and report as necessary.

## 4. Conclusions

Löwe syndrome is a rare disease that still poses diagnostic and, especially, treatment challenges. These individuals could benefit from effective management and an improvement in their quality of life if the disease was promptly identified and confirmed through molecular genetic analysis. RhGH therapy could be a valuable part of the active treatment, though its duration and final outcome are still uncertain.

## Figures and Tables

**Figure 1 children-10-01166-f001:**
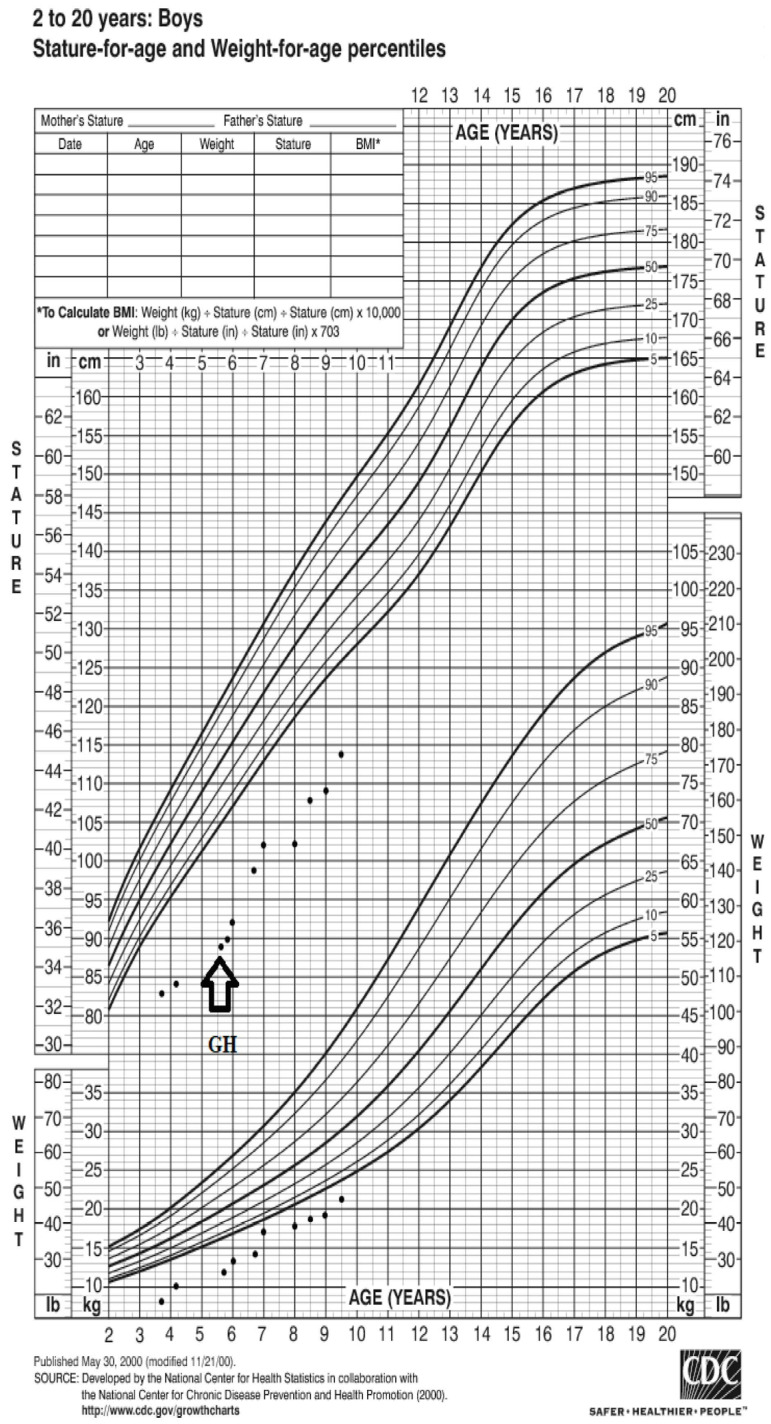
Growth chart of the patient (www.cdc.gov/growthcharts, accessed on 1 July 2023).

**Figure 2 children-10-01166-f002:**
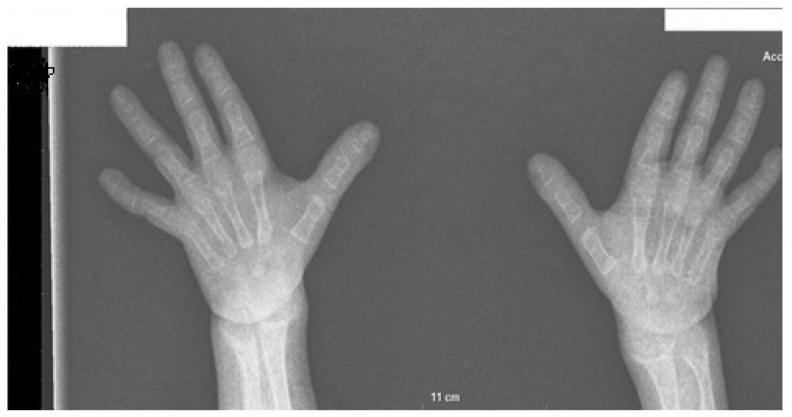
Radiogram of the wrists of the patient at 3 years 7 months of age showing severe rickets. Diffusely reduced bone mineralization, widened metaphyseal surfaces of the distal radius, ulna, and metacarpal bones with uneven contours to the epiphyseal plates are evident, as well as expanded growth plates (cupping) with soft tissue thickening of the wrist and distal metacarpal bones. Bone age measured by the Greulich and Pyle atlas method corresponds to that of a 9-month-old boy and is more than 2SD behind the chronological age.

**Figure 3 children-10-01166-f003:**
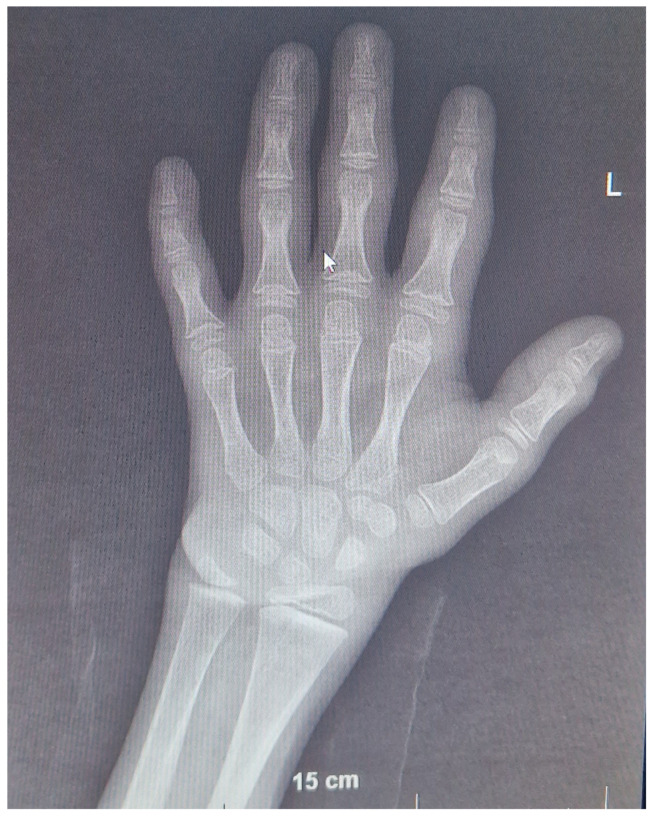
Radiogram of the patient’s wrists at 9 years 6 months of age. Bone age corresponds to 8 years 6 months according to the Greulich and Pyle method, which is less than −2 SD behind the chronological age. No evidence of rickets was found.

**Table 1 children-10-01166-t001:** Dynamics of the laboratory values with time.

	At Presentation (3 Years 7 Months)	At 4 Years 10 Months	At GH Start (5 Years 6 Months)	1 Year after GH Start (6 Years 5 Months)	At 7 Years	At 8 Years 7 Months	At 9 Years 6 Months
pH (7.34–7.44)	7.33	7.41	7.43	7.35	7.34	7.35	7.45
pCO_2_ kPa (4.66–5.99)	3.01	4.15	4.43	4.26	3.50	4.30	5.1
AB mmol/L (21.0–25.0)	11.7	15.0	13.0	17.5	18.0	17.7	23.5
BE mmol/L (−2.5–2.5)	−11.9	−4.4	−1.7	−7.9	−7.0	−7.9	−1.1
K mmol/L (3.5–6.0)	3.4	5.0	4.7	4.0	3.6	3.9	4.3
Ca mmol/L (2.18–2.60)	2.26	2.70	2.80	2.39	2.49	2.46	2.46
P mmol/L (1.29–2.26)	0.5	1.4	1.8	1.2	-	1.2	1.16
TSH uIU/mL (0.4–4.0)	2.81	-	-	5.06	-	3.71	2.96
FT4 pmol/L (10.3–24)	11.20	15.70	16.60	16.90	11.92	14.75	17.22
Urea (BUN) mmol/L (3.2–8.2)	4.1	7.9	6.2	4.3	6.1	9.5	6.5
Creatinine μmol/L (18–62)	34	47	66	61	63	70	71
Intact PTH pg/mL (11–87)	89.9	<3.0	<3.0	14.0	-	54.7	6.5
IGF-1 ng/mL (47–231)	<25.0	29.0	37.0	67.0	135.5	93.0	119
rhGH (mg/kg/day)	-	-	0.023	0.035	0.045	0.055	0.055
25-OH-Vit. D ng/mL (9.30–47.90)	19.80	-	-	25.40	-	22.94	21.06

AB—actual bicarbonate; BE—base excess; K—potassium; Ca—calcium; P—phosphorus; TSH—Thyroid-Stimulating Hormone; PTH—parathyroid hormone; IGF-1—Insulin-like growth factor 1; rhGH—recombinant human growth hormone; 25-OH-Vit. D—25-hydroxy vitamin D.

## Data Availability

Not applicable.

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
