# Peer review of "Initial Effect of Recombinant Human Growth Hormone Treatment in a Patient with Löwe Syndrome"

_children, 2023, doi:10.3390/children10071166_

Round 1

Reviewer 1 Report

In "Initial effect of recombinant..", Iotova et al showed a case report of a patient with Lowe syndrome treated with human Growth Hormone.  The case is interesting and is reportable.  There are writing issues.

Abstract: "impaired physical and mental development" is not accurate.  No "the" in the first mention of oculocerebrorenal syndrome.  Congenital cataracts, secondary glaucoma should be mentioned. Mayo Clinic  should be omitted; not necessary in abstracts.  Overall it should be re-written for English-proof.

Overall the references are quite lacking.  Over 350 publications on OCRL and Lowe syndrome are present, yet only 6 references are here.

For the growth-chart, the authors should compare with the compiled Growth Chart for 137 Lowe children in (PMID: 35803701  PMCID: PMC10186212).

Figure 2-3 should have more detailed description of the X-rays, and mark the defects on the X-ray for general audience.

Abstract: "impaired physical and mental development" is not accurate.  No "the" in the first mention of oculocerebrorenal syndrome.  Congenital cataracts, secondary glaucoma should be mentioned. Mayo Clinic  should be omitted; not necessary in abstracts.  Overall it should be re-written for English-proof.

Overall, the writing is not for medical publication and should be more descriptive but less colloquial 

He couldn’t stand alone or even with help and couldn’t walk."

"did not control urination and defecation" are some examples.

Author Response

Dear editorial team, dear reviewers,

Thank you for your letter. We appreciate the interest that the editors and reviewers have taken in our manuscript and the constructive criticism they have given.

Please consider our revised manuscript, “Initial effect of recombinant human growth hormone treatment in a patient with Löwe syndrome" by Iotova et al., children-2473685, for publication in “Children”. 

We have addressed the major concerns of the reviewers and we submit a list of changes. We increased the number of words according to suggestion of the editor. To modify and improve the English style, the entire article was rewritten. Also, the changes of the manuscript were made by using tracked changes.

Thank you again for consideration of our revised manuscript.

With best regards,

The authors

Reviewer 1

In "Initial effect of recombinant..", Iotova et al showed a case report of a patient with Lowe syndrome treated with human Growth Hormone.  The case is interesting and is reportable.  There are writing issues.

 Thank you for your comment.

Abstract: "impaired physical and mental development" is not accurate.  No "the" in the first mention of oculocerebrorenal syndrome.  Congenital cataracts, secondary glaucoma should be mentioned. Mayo Clinic should be omitted; not necessary in abstracts.  Overall it should be re-written for English-proof.

Thank you for your suggestion. We added the above mentioned symptoms to the description of the case and rewrote the whole manuscript.

Overall the references are quite lacking.  Over 350 publications on OCRL and Lowe syndrome are present, yet only 6 references are here.

Thank you for your comment. We added additional references to the list.

For the growth-chart, the authors should compare with the compiled Growth Chart for 137 Lowe children in (PMID: 35803701  PMCID: PMC10186212).

Thank you for this suggestion. In practice, the Growth Chart of our patient resembles the published charts.

Figure 2-3 should have more detailed description of the X-rays, and mark the defects on the X-ray for general audience.

Thank you for your suggestion. We added more information to the figure legends.

Comments on the Quality of English Language

Abstract: "impaired physical and mental development" is not accurate.  No "the" in the first mention of oculocerebrorenal syndrome.  Congenital cataracts, secondary glaucoma should be mentioned. Mayo Clinic  should be omitted; not necessary in abstracts.  Overall it should be re-written for English-proof.

Overall, the writing is not for medical publication and should be more descriptive but less colloquial

He couldn’t stand alone or even with help and couldn’t walk."

"did not control urination and defecation" are some examples.

Thank you for your comment. We revised the English language of the whole manuscript.

Reviewer 2 Report

This is a well written case report of Lowe syndrome describing use of recombinant growth hormone therapy.

Minor revisions will improve the case report.

Introduction – OCRL is expressed all over the body and likely has a role in multiple functions. It is incorrect to assume that most of the clinical manifestations are due to renal dysfunction.

The references to growth related SD should include the reference growth chart, e.g. based on CDC 2000 growth charts.

There are likely multiple reasons for improvement of health for this child. It is important to note that the improvements are from several aspects of treatment, not solely from recombinant growth hormone therapy. E.g. improvements in kidney disease and cryptorchidism may not be linked to growth hormone therapy and should be clarified as such.

Please describe the seizures and note whether or not EEG was done.  

English language editing, especially limiting the use of upper case letters in the middle of the sentences.

Please make sure to italicize human gene notation to align with convention.

Author Response

Dear editorial team, dear reviewers,

Thank you for your letter. We appreciate the interest that the editors and reviewers have taken in our manuscript and the constructive criticism they have given.

Please consider our revised manuscript, “Initial effect of recombinant human growth hormone treatment in a patient with Löwe syndrome" by Iotova et al., children-2473685, for publication in “Children”. 

We have addressed the major concerns of the reviewers and we submit a list of changes. We increased the number of words according to suggestion of the editor. To modify and improve the English style, the entire article was rewritten. Also, the changes of the manuscript were made by using tracked changes.

Thank you again for consideration of our revised manuscript.

With best regards,

The authors

Reviewer 2

This is a well written case report of Lowe syndrome describing use of recombinant growth hormone therapy. Minor revisions will improve the case report.

Thank you for your comment.

Introduction – OCRL is expressed all over the body and likely has a role in multiple functions. It is incorrect to assume that most of the clinical manifestations are due to renal dysfunction.

Thank you for your comment. We added the following new information, regarding the OCRL gene: The OCRL gene codes for the protein OCRL-1, which is a phosphatase enzyme. This enzyme is involved in cell transport and in maintaining the integrity of the cell by regulating the actin cytoskeleton.

The references to growth related SD should include the reference growth chart, e.g. based on CDC 2000 growth charts.

Thank you for pointing at this omission. It is now corrected in the text.

There are likely multiple reasons for improvement of health for this child. It is important to note that the improvements are from several aspects of treatment, not solely from recombinant growth hormone therapy. E.g. improvements in kidney disease and cryptorchidism may not be linked to growth hormone therapy and should be clarified as such.

Thank you for your comment. We included more information in the Discussion section, paragraph 6 and added information about the various aspects of the improvement of the patient’s state.

Please describe the seizures and note whether or not EEG was done. 

Thank you for this valuable suggestion. This was now done in the manuscript.

Comments on the Quality of English Language

English language editing, especially limiting the use of upper case letters in the middle of the sentences.

Thank you for your comment. We revised the whole manuscript.

Please make sure to italicize human gene notation to align with convention.

Thank you for your comment. We italicized the names of the genes mentioned in the text.